# Does Geometric Structure in Convolutional Filter Space Provide Filter Redundancy Information?

**Anshul Thakur**               ANSHUL.THAKUR@ENG.OX.AC.UK
*Dept. of Engineering Science, University of Oxford, UK*

**Vinayak Abrol**                      ABROL@IIITD.AC.IN
*Infosys Centre for AI, IIIT Delhi, India*

**Pulkit Sharma**             PULKITSHARMAY2K@GMAIL.COM
*Dept. of Engineering Science, University of Oxford, UK*

**Editors:** Sophia Sanborn, Christian Shewmake, Simone Azeglio, Arianna Di Bernardo, Nina Miolane

## Abstract

This paper aims to study the geometrical structure present in a CNN filter space for investigating redundancy or importance of an individual filter. In particular, this paper analyses the convolutional layer filter space using simplical geometry to establish a relation between filter relevance and their location on the simplex. Convex combination of extremal points of a simplex can span the entire volume of the simplex. As a result, these points are inherently the most relevant components. Based on this principle, we hypothesise a notion that *filters lying near these extremal points of a simplex modelling the filter space are least redundant filters and vice-versa.* We validate this positional relevance hypothesis by successfully employing it for data-independent filter ranking and artificial filter fabrication in trained convolutional neural networks. The empirical analysis on different CNN architectures such as ResNet-50 and VGG-16 provide strong evidence in favour of the postulated positional relevance hypothesis.

**Keywords:** CNN filter redundancy, Positional filter relevance, Geometric modelling

## 1. Introduction

Deep convolution neural networks (CNNs) have exhibited state-of-the-art performance in different pattern analysis tasks LeCun et al. (2015); He et al. (2016). This success inspired thorough investigations in inner working of CNNs resulting in our current understanding of the presence of massive filter redundancy in over-parameterised CNNs. Many studies have targeted this issue by pruning CNNs to obtain compact and efficient models for resource constrained environment. A standard approach to prune is by removing redundant or less relevant filters from convolutional layers. The existing studies determine filter relevance rankings either by exploiting training data or by analysing the filters themselves in a data-independent manner Li et al. (2017); He et al. (2018). The data-dependent methods determine the filters' contribution by studying the activation maps or by analysing the impact of filter removal on the empirical loss Ding et al. (2019); Sándor et al. (2020). While data-independent filter ranking strategies are expected to be sub-par against their data-dependent counterparts, they have a potential to improve our understanding of convolutional filter spaces.

Inspired by the data-independent filter ranking methods, this paper studies the geometric structure of convolutional filter space of a CNN layer to establish filter relevance based

Input image 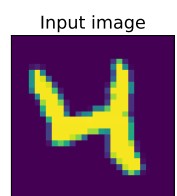

Activation by vertical Prewitt operator 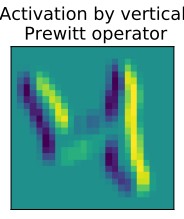

Activation by horizontal Prewitt operator 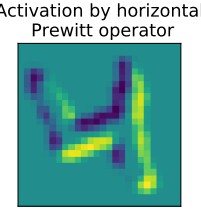

Activation by synthesized operator 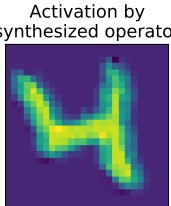

Figure 1: Illustration of activation maps generated by applying Prewitt operators (vertical and horizontal) and a convolutional filter synthesised by a convex combination of Prewitt operators.

on their position or location in the space. To this aim, we model convolution filters at a CNN layer using simplical geometry and convex combination. Based on this modelling, we postulate the following:

- A filter represented as a convex combination of two filters can be seen as redundant and the information loss procured by its removal can be compensated by other filters in the combination. This behaviour is illustrated in Figure 1. A convex combination of horizontal and vertical Prewitt edge detection operators Prewitt (1970) is used to synthesise a new operator as: $\mathbf{z} = 0.5 \times \mathbf{x} + 0.5 \times \mathbf{y}$, where $\mathbf{x}$ and $\mathbf{y}$ are Prewitt operators. These three operators are used as filters in a convolution layer (with no bias and linear activation), and activation maps are obtained for an input image. The visual inspection of these activation maps shows that filter $\mathbf{z}$ detects both horizontal and vertical edges. However, it doesn't capture any specific structural information that is not already present in either of the activation maps corresponding to filters $\mathbf{x}$ and $\mathbf{y}$.

- If we fit a simplex[1] over any convolutional layer filters, then the convex combination of extremal points or vertices of this simplex can model all the respective filters. These extremal points are unique and act as *bases* of the filter space where finite linear combination of bases is constrained to be the convex. Hence, these extremal points have more relevance than filters lying inside the simplex. Consequently, we can arrive on a *relation* between filter relevance and their location with respect to extremal points.

In this work, we use archetypal analysis Cutler and Breiman (1994); Chen et al. (2014); Abrol and Sharma (2020), a matrix factorization framework, to model filter space using a simplex. The archetypes approximate the extremal points and hence, the convex hull of the simplex. This paper utilises these approximated extremal points or archetypes and principles of simplical geometry to quantify relevance or redundancy of the filters. Filters lying near the geometric median or centroid of the filter space, and hence the simplex, can be modelled as convex combination of archetypes and can be considered as redundant. Based on these concepts, this paper formally conceptualise a filter ranking method (described in Section 4) that can be used for successful CNN filter pruning.

---

1. A simplex is a generalization of triangle or tetrahedron to arbitrary dimensions.

Apart from filter ranking, the positional relevance and simplical geometry can be used to fabricate new convolution filters. Since archetypes are considered as basic blocks of a filter space, their random convex combinations can result in fabricated filters lying in the same input filter space. In a transfer learning paradigm, these "fabricated filters" can be used to decrease the storage footprints of a pre-trained CNN. Rather than storing the entire models, only archetypes can be stored for different convolution layers and filters for each layer can be generated on the fly by random convex combinations. Since fabricated filters lie in the span of original filters, the fabricated filters also provide an effective initialization for transfer learning.

The main contributions of this study are listed below:

- This paper utilises simplical geometry to present a relation between filter relevance and their location in filter space with respect to the extremal points of the simplex.
- The location of filters and principles of simplical geometry are used to propose a data-independent filter ranking algorithm and an artificial filter fabrication method for decreasing storage foot-prints of large pre-trained models in transfer learning.

The rest of this paper is organised as follows. Section 2 discusses the existing data-independent filter analysis studies and their similarities to the proposed simplical filter analysis. In Section 3, archetypal analysis and simplical modelling of convolutional filters are discussed. The proposed data-independent filter ranking framework and convolution filter fabrication are presented in Section 4 and Section 5, respectively. Experimental setup and results are discussed in Section 6 and Section 7, respectively. Finally, Section 8 concludes this paper.

## 2. Existing data-independent filter ranking methods

Data-independent convolutional filter analysis is a sparsely studied area of deep learning. Most of data-independent filter analysis utilise norm-based filter ranking for CNN pruning Li et al. (2017); He et al. (2018). The lower norm filters lead to weaker activations which often point to the lesser contribution. These studies analyse filters individually and do not consider any relation among filters. These methods ignore the fact that high norm filters can also be redundant with respect to each other i.e. they are identical or generate near identical filter response. Hence, most of the redundant filters are not removed from this analysis.

In He et al. (2019a), the authors highlighted this drawback of norm-based analysis and explored similarity among geometric median and neighbouring filters to define redundancy. They remove filters that are similar to or lie near the geometric median. The intuition behind this approach is that the remaining filters can compensate for the contribution of removed filters. However, this study does not elaborate on the choice of the geometric median. In theory, any filter can be used as an anchor for removing the neighbouring filters. In He et al. (2019b), the authors proposed using both filter norms and relations among filters for ranking. This study quantifies the relation among filters by computing a filter's distance from every other filter. These distances are accumulated to form a ranking metric. The higher accumulated distance of a filter reflects its low correlation with other filters and uniqueness.

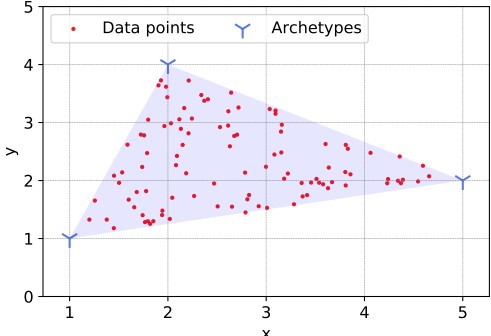

Figure 2: Illustration of a 2-simplex whose vertices are defined by 3 archetypes computed from data points. The shaded region represents the span of convex combination of vertices of the simplex.

The proposed simplical geometry-based filter analysis has some similarity with geometric median-based filter ranking (GMFR) He et al. (2019a). Both these methods utilise the location of filters with respect to a geometric structure to decide the relevance of filters. However, GMFR doesn't provide any explanation for redundancy near the geometric median. Moreover, it does not provide any mechanism to determine the most relevant filters. The proposed geometrical analysis overcomes these drawbacks and provides an explanation for the intuition of redundancy utilised in GMFR.

## 3. Simplical modelling of convolution filters

This section presents the simplical modelling of convolutional filters of a CNN layer using archetypal analysis.

### 3.1. Symbols and pre-processing

Each CNN is assumed to have $L$ layers, and the connection between $(i-1)$th and $i$th convolutional layers can be represented by $\mathbf{W} \in \mathbb{R}^{K \times K \times N_{i-1} \times N_i}$. Here $K \times K$ is the kernel size of convolutional filters at $i$th layer, $N_{i-1}$ is the number of input channels or number of filters at $(i-1)$th layer and $N_i$ is the number of filters at $i$th layer. Extrapolating this notation, a $j$th convolutional filter at $i$th layer can be represented as: $\mathbf{F}_{i,j} \in \mathbb{R}^{K \times K \times N_{i-1}}$.

To simplify the computation, each convolutional filter is vectorized. As a result of this transformation, the convolutional filters at $i$th layer are now represented by a 2D matrix, $\mathbf{X} \in \mathbb{R}^{l \times N_i}$ where each column contains $l$-dimensional filter ($l = K \times K \times N_{i-1}$).

### 3.2. Archetypal analysis

Archetypal analysis (AA) is a matrix decomposition method that factorises a matrix into archetypes and convex representations. Given an input matrix $\mathbf{X} \in \mathbb{R}^{l \times N_i}$ (each column represents a data point or filter), AA decomposes $\mathbf{X}$ as $\mathbf{X} \approx \mathbf{DA}$ where $\mathbf{D} \in \mathbb{R}^{l \times d}$ is an archetypal dictionary containing $d$ archetypes. Also, $\mathbf{A} \in \mathbb{R}^{d \times N_i}$ is a convex representation

matrix. Hence, each data point in $\mathbf{X}$ is represented as the convex combination of archetypes. Additionally, archetypes are also restricted to be the convex combination of data points i.e. $\mathbf{D} = \mathbf{XB}$ where $\mathbf{B} \in \mathbb{R}^{N_i \times d}$ is a convex representation matrix. Both these conditions force the archetypes to lie on the convex hull of the data points.

The archetypal dictionary $\mathbf{D}$ can be obtained by solving the following least-squares optimization problem with convex constraints Abrol and Sharma (2020):

$$\underset{\substack{\mathbf{B},\mathbf{A} \\ \mathbf{b}_j \in \Delta_{N_i}, \mathbf{a}_i \in \Delta_d}}{\operatorname{argmin}} \quad \|\mathbf{X} - \mathbf{XBA}\|_F^2, \tag{1}$$
$$\Delta_{N_i} \triangleq [\mathbf{b}_j \succeq 0, \|\mathbf{b}_j\|_1 = 1], \Delta_d \triangleq [\mathbf{a}_i \succeq 0, \|\mathbf{a}_i\|_1 = 1].$$

Here $\mathbf{a}_i$ and $\mathbf{b}_j$ are columns of $\mathbf{A} \in \mathbb{R}^{d \times N_i}$ and $\mathbf{B} \in \mathbb{R}^{N_i \times d}$, respectively. Eq. 1 can be solved using a block-coordinate descent algorithm proposed by Chen *et al.* Chen et al. (2014).

Since archetypes lie on the convex hull and their convex combinations can model the span of input data points or filters, archetypes can be seen as approximations to the extremal points or vertices of the simplex. Hence, using $d$ archetypes, we fit a $(d-1)$-simplex[2] over the input filters where archetypes define vertices of this simplex. Figure 2 illustrates a 2-simplex obtained by 3 archetypes computed from 2D data points by solving Eq. 1.

## 4. Filter Ranking Based on Positional Relevance Hypothesis

As per the hypothesized positional relevance (Section 1), filters lying near vertices of the simplex or near archetypes (hence, convex hull) modelling the filter space are considered as less redundant or unique. Whereas, filters that can be represented as convex combination of these archetypes are considered redundant. Based on this hypothesis, we can build a quantifiable filter ranking mechanism.

As discussed in Section 3.2, a matrix $\mathbf{X} \in \mathbb{R}^{l \times N_i}$ whose columns contain $N_i$ filters is factorised into a dictionary containing $d$ archetypes, $\mathbf{D} \in \mathbb{R}^{l \times d}$, and convex representation matrix, $\mathbf{A} \in \mathbb{R}^{d \times N_i}$. The $k$th column of $\mathbf{A}$ ($\mathbf{a}_k$) is a convex representation vector whose elements represent the contribution of each archetype in defining the $k$th filter or $k$th column of $\mathbf{X}$. We rely on sparsity of these convex representation vectors for filter ranking. The filters having higher $\ell_0$-norm of convex representation vectors are considered as redundant and are ranked low. We can obtain the filter ranking by sorting them based on $\ell_0$-norm of convex representations.

Suppose all archetypes have an equal contribution in defining a filter. As a result, the corresponding convex representation vector exhibits the highest possible $\ell_0$-norm. Geometrically, this filter is equidistant from all archetypes and lie at the centroid of the simplex. Extending this argument, a filter lying in the vicinity of an archetype has a significant contribution from this archetype, and its convex representation vector exhibits maximum possible sparsity. Hence, the sparsity or $\ell_0$-norm of convex representation vectors obtained during AA is a viable way to get an estimate of filter location and hence, their positional relevance.

---

2. A $(d-1)$-simplex has $d$ vertices.

## 5. Artificial Filter Fabrication

Given archetypes obtained at a convolutional layer, filters can be fabricated by computing the random convex combinations of these archetypes:

$$\mathbf{C} = \mathbf{D}\mathbf{Y}^T \ \text{ where } \ \mathbf{y}_{[i]} \succeq 0 \ \text{ and } \ \|\mathbf{y}_{[i]}\|_1 = 1. \tag{2}$$

Here $\mathbf{D} \in \mathbb{R}^{l \times d}$ is a dictionary of archetypes, $\mathbf{Y} \in \mathbb{R}^{N \times d}$ is randomly sampled matrix where each row $\mathbf{y}_{[i]}$ follow the constraints of a convex representation and $\mathbf{C} \in \mathbb{R}^{l \times N}$ contains $N$ fabricated convolutional filters. $\mathbf{C}$ can be reshaped to undone the pre-processing to obtain new weight connections, $\mathbf{W}' \in \mathbb{R}^{K \times K \times N_{i-1} \times N}$.

The span of the convex combination of archetypes includes the input filters. Hence, fabricated filters and the "original" filters are expected to exhibit similar behaviour. The main application of the fabricated filters lies in transfer learning paradigm to decrease the memory footprint of a pre-trained model. Instead of storing or transferring the entire pre-trained models, only layer-specific archetypes are required. The convolutional filters can be fabricated using these archetypes on-the-fly. Since archetypes are fewer than the overall number of filters, this process decreases the models' storage footprints without reducing the model capacity.

## 6. Experimental Setup

We designed three experiments to provide empirical evidence in favour of the positional relevance hypothesis:

- We remove fixed number of filters from the trained CNNs (*LeNet-5* LeCun et al. (1998) trained on MNIST dataset and larger CNNs such as *VGG16* and *ResNet-50* trained on CIFAR-10 dataset Krizhevsky (2009)) based on their position on the simplex or filter space and analyse the drop in performance. Filters are removed either from *near geometric median* locations or *near archetypal* or *near convex hull* locations.

- We perform artificial filter fabrication using archetypes at each conv layer of *VGG-16* and *ResNet-50* (trained on ImageNet). We compare the performance of artificial filters and original filters as model initialisation for training on CIFAR-10 within transfer learning paradigm.

- We prune *VGG-16* and *ResNet-50* (trained on CIFAR-10) using the proposed filter ranking mechanism at 50% and 75% pruning rates (percentage of filters removed at each layer). Pruning consists of removing filters and re-training the models. The performance is compared against random filter removal and the common data-independent filter ranking methods (discussed in Section 2.

***Parameter setting:*** In first experiment, we remove 5%, 10%, 15% and 20% filters from each CNN layer without re-training. Both for filter ranking and artificial filter fabrication, the number of archetypes learned at each layer is fixed to be the 25% of the total filters. For fine-tuning on artificial filters or re-training after pruning, we used 50 training epochs, a batch-size of 64 and Adam optimiser with a fixed learning rate of 0.0001. Each experiment is performed 10 times and average performance across 10 runs is reported.

| Model | Baseline | Filters removed at each layer (%) | | | | | | | |
|---|---|---|---|---|---|---|---|---|---|
| | | 5% | | 10% | | 15% | | 20% | |
| | | Near archetypes | Near geometric median | Near archetypes | Near geometric median | Near archetypes | Near geometric median | Near archetypes | Near geometric median |
| ResNet-50 (CIFAR-10) | 95.9 | 89.1 | **94.3** | 84.5 | **90.38** | 65.3 | **73.29** | 51.9 | **67.8** |
| VGG16 (CIFAR-10) | 93.5 | 91 | **93.1** | 87 | **91.7** | 79.4 | **90.2** | 64.6 | **84.1** |
| LeNet-5 (MNIST) | 99.2 | 98.8 | **99.18** | 98.5 | **99.15** | 98.6 | **99.1** | 97.2 | **98.9** |

Table 1: Effect of location of removed filters on the classification accuracy of CNNs.

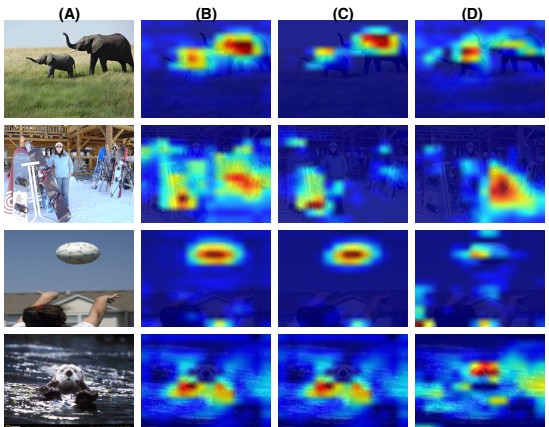

Figure 3: Grad-CAM visualisation for *VGG16* pre-trained on ImageNet. Column (A) contains the input examples. Column (B) shows the standard Grad-CAMs. Columns (C) and (D) contain the Grad-CAMs obtained after removing 384 "near geometric median" and "near archetypal" filters from the last convolution layer, respectively.

**Implementation details:** All the experiments are conducted using Python and Tensorflow. To implement archetypal analysis (Equation 1), we have used *Coreset based archetypal analysis* Mair and Brefeld (2019).

## 7. Results & Discussion

### 7.1. Impact of position of removed filters on CNNs

The results of this analysis are tabulated in Table 1. Across all configurations, it is clear that removing filters lying near the geometric median leads to lesser performance degradation than removing filters lying near archetypes. This provides a strong empirical evidence in favour of positional relevance hypothesis.

To visually compare the sensitivity of locations of filters, we compute gradient weighted class activation maps (Grad-CAM) Selvaraju et al. (2017) for four random ImageNet examples using *VGG16* where last convolution layer has 512 filters. Three variants of Grad-CAM are computed: a standard Grad-CAM using all filters, Grad-CAM after removing 384 (i.e.

| Models | Initialization | | |
|---|---|---|---|
| | Random | Imagenet weights | Fabricated filters |
| ResNet-50 | 95.9 ($\pm$ 0.09) | 97.2 ($\pm$ 0.1) | **97.5** ($\pm$ **0.12**) |
| VGG16 | 93.1 ($\pm$ 0.12) | **94.9** ($\pm$ **0.18**) | 94.3 ($\pm$ 0.2) |

Table 2: Performance of fabricated filters as initialised weights in the transfer learning paradigm. Imagenet weights are used for filter fabrication and all CNNs are evaluated on *CIFAR-10* dataset.

| Pruning Rate (%) | Performance on VGG-16 (*Baseline: 93.1 %*) | | | Performance on ResNet-50 (*Baseline: 95.96%*) | | |
|---|---|---|---|---|---|---|
| | $\ell_1$-norm Li et al. (2017) | GMFR He et al. (2019a) | Positional Relevance | $\ell_1$-norm Li et al. (2017) | GMFR He et al. (2019a) | Positional Relevance |
| 50 | 88.15 ($\pm$ 0.35) | 88.79 ($\pm$ 0.21) | **88.94** ($\pm$ **0.19**) | 93.75 ($\pm$ 0.21) | 94.75 ($\pm$ 0.15) | **94.85** ($\pm$ **0.28**) |
| 75 | 83.7 ($\pm$ 0.54) | **84.15** ($\pm$ **0.35**) | 84.1 ($\pm$ 0.3) | 92.91($\pm$ 0.37) | 93.18 ($\pm$ 0.29) | **93.25** ($\pm$ **0.27**) |

Table 3: Performance of positional relevance based filter ranking in a pruning framework.

75% of filters) "near geometric median" filters and Grad-CAM after removing 384 "near archetypes" filters. Figure 3 illustrates the difference in these three variants of Grad-CAM. The analysis of this figure illustrates that removing "near archetypes" filters leads to more deviation in the most salient regions (identified by the standard Grad-CAM) as compared to the removal of "near geometric median" filters. This observation further enforces that filters lying near convex hull have more relevance than filters lying near the geometric median.

## 7.2. Performance of fabricated filters

The models are initialised with random filters, original ImageNet weights/filters and fabricated filters. These models are fine-tuned, and their performance is compared. The results of this experiment are reported in Table 2. It can be observed that models with fabricated filters led to a significant improvement in classification performance over the random initialisation while achieving comparable performance to the pre-trained models. This corroborates the claim that fabricated filters are informative, and are indeed a viable option for reducing the storage footprints of the pre-trained CNNs. By only storing archetypes (that are 25% of the filters at a layer), we are able to reduce the storage footprints of *ResNet-50* and *VGG16* by 65.25% (from 99 MB to 34.4 MB) and 7.38% (from 528 MB to 489 MB), respectively.

## 7.3. Pruning ResNet-50 and VGG16

Table 3 reports the performance of the data independent filter ranking for pruning *VGG-16* and *ResNet-50* trained on *CIFAR-10*. The following observations can be drawn from the analysis of this figure:

- The proposed positional relevance based filter ranking either outperforms or shows comparable performance against other comparative methods on both datasets, irrespective of the pruning rate. This success in pruning further affirms the positional relevance hypothesis.

- $\ell_1$-*norm* based method removes a significant number of the relevant filters, and hence, are consistently outperformed by GMFR and the proposed method. Instead of weaker filter activations, both of these methods exploit the redundancy among filters to obtain filter ranking for pruning.

- The performance of the proposed positional relevance based filter ranking and any data-independent methods is not comparable against the data-dependent pruning methods. The purpose of this study was not to come up with a best pruning strategy to study and establish the positional relevance in a CNN filter space.

## 8. Conclusion

This paper explored the simplical geometry to present a relation between the filter relevance and the location of a filter with respect to the simplex or the convex hull of filters at a layer. This paper experimentally verified that the filters lying on the convex hull (i.e. the extremal filters) are unique and are more relevant in the decision-making process. The notion of positional relevance is further corroborated by proposing a data-independent filter ranking method and filter fabrication method. The performance of the proposed filter ranking method in a typical CNN pruning framework highlights that the hypothesis of positional relevance holds in some of the well-known CNNs (that are considered for this study). Moreover, the performance of fabricated filters in the transfer learning paradigm again corroborates that the extremal filters are highly informative, and their convex combination can model the entire filter space. Future work may deal with exploring extremal points of the simplex to understand the relevant characteristics or objects learnt at each convolutional layer of a CNN.

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

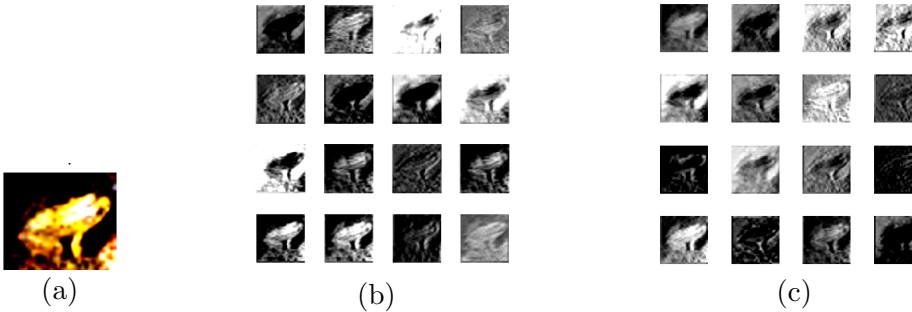

(a)    (b)    (c)

Figure 4: Comparison between the activation maps obtained from (b) trained convolution filters and (c) fabricated filters at the first layer of *VGG-16* (trained on *CIFAR-10*) for an input example (a).

## Appendix A. Activation maps from fabricated filters

We generated fabricated filters from the first layer filters of a trained *VGG16* model. Figure 4 shows the activation maps generated from the fabricated filters and the original filters. This figure suggests that both sets of filters are capturing similar information (i.e. edges and texture).

