# OpenReview forum: "Does Geometric Structure in Convolutional Filter Space Provide Filter Redundancy Information?"
_NeurIPS.cc/2022/Workshop/NeurReps — NeurReps 2022 Poster_

### Official Review · Reviewer_aYnp · 2022-10-09
**Simplical Geometry to Understand Filter Relevance**

**Confidence:** 4
**Soundness:** 4
**Presentation:** 3
**Contribution:** 3
**Overall Rating:** 7

**Summary:**

The authors propose a *positional relevance hypothesis* which gives a data-independent ranking of the relevance of the CNN filters in a given layer. The hypothesis states that filters which lie near extremal points of the simplex containing filters in a layer are most relevant since all other filters can be constructed as convex combinations of these extremal points. Conversely, filters inside of the simplex are hypothesized to be less relevant.

To test this idea, the authors provide an algorithm which identifies a collection of archetypal filters, from which all other filters can be reconstructed through convex combinations. In pre-trained CNNs, they find that removing filters closer to the archetypes leads to larger degradation in performance than removing filters near the geometric median. Similarly, if networks are retrained after pruning, their performance degrades more if the extremal filters are removed. This technique gives competitive performance to other data-independent relevance ranking techniques. Lastly, they show that initializing a network with filters constructed with convex combinations of the extremal filters can give a slight improvement over random initialization on CIFAR-10.


**Questions:**

I had a few questions about the work which could improve my understanding and maybe sharpen the message of the paper.

How are extremal points are *mechanically* the most important for network performance, in particular why is it the case that interior filters in the simplex can be removed with little degradation in performance but removing extremal points reduces test accuracy?

Is the positional relevance hypothesis a property only of networks trained with gradient descent? Could someone randomly initialize a CNN, identify a set of archetypes with the proposed algorithm (without training), and then prune to keep these and expect better performance after subsequent training? The discussion in the paper made it seem like the logic should also apply to untrained CNNs as well.

Do the authors have any thoughts about how the geometry of the filters relates or corresponds to the activation patterns in the network and the network predictions?


**Limitations:**

The authors were upfront about the limitations of their work. They clearly state that they are not trying to provide a state of the art pruning method, but rather aim to characterize how geometric properties of filter space can be used to assess relevance.

**Recommended Decision:**

3: Accept

**Relevance:**

4: Highly relevant

**Strengths And Weaknesses:**

Strengths:
This work provides a novel geometric perspective on filter relevance ranking which has empirical support and could provide inspiration for future studies on this topic. The empirical analysis and archetype discovery methods all appeared sound on my reading. The authors do a good job comparing to relevant baselines (l1 ranking of Li et al and the GMFR of He et al). The writing is clear figures and tables are easy to read.

Weaknesses:

While this work's strengths are apparent, the paper did have a weakness with respect to the scope of the hypothesis and providing a proposed mechanism of this effect. Specifically, the authors do not make clear under what conditions they expect this to hold and whether or not this property is unique to fully trained models (see Questions below).

Minor things
Maybe consider presenting the pruning + retraining results (Table 3) after the pruning without retraining (Table 1) rather than separating these results with the different initialization schemes (Table 2) in between.

**Submission Track:**

Proceedings Paper (9 Page)

---

### Official Review · Reviewer_2LVR · 2022-10-16
**Does Geometric Structure in Convolutional Filter Space Provide Filter Redundancy Information?**

**Confidence:** 4
**Soundness:** 3
**Presentation:** 3
**Contribution:** 2
**Overall Rating:** 6

**Summary:**

In order to determine if a certain filter is redundant, this research examines the geometrical structure found in the CNN filter space. To establish a link between filter significance and a filter's position on the simplex, this work examines the convolutional layer filter space in particular utilizing simplical geometry. Authors hypothesize that filters that are closest to these extremal points of a simplex model of the filter space . By successfully using it for data-independent filter ranking and artificial filter creation in trained convolutional neural networks, authors support this positional relevance hypothesis. The empirical investigation of various CNN architectures, including ResNet-50 and VGG-16, provide compelling support for the hypothesised positional relevance.



**Questions:**

Clarify with more technical details methods developed by Cutler and Breiman (1994); Chen et al. (2014); Abrol and Sharma (2020), about a matrix factorization framework, to model filter space using a simplex.

**Limitations:**

Add a state-of-the-art on:
-  the simplical geometry method used for this applications
- alternative methods for  studying redundancy of an individual filter

**Recommended Decision:**

2: Borderline

**Relevance:**

2: Limited relevance

**Strengths And Weaknesses:**

Strenghts:
Authors designed three experiments to provide empirical evidence in favour of the positional relevance hypothesis
=> Results are illustrated on existing data sets.

Weaknesses:
Authors uses Cutler and Breiman (1994); Chen et al. (2014); Abrol and Sharma (2020), a matrix factorization framework, to model filter space using a simplex. They utilises these approximated extremal points or archetypes and principles of simplical geometry to quantify relevance or redundancy of the filters.
=> these tools are not enough technically explained

**Submission Track:**

Proceedings Paper (9 Page)

---

### Official Review · Reviewer_n5no · 2022-10-16
**Interesting paper, but lacking in depth**

**Confidence:** 4
**Soundness:** 3
**Presentation:** 3
**Contribution:** 2
**Overall Rating:** 5

**Summary:**

__Summary.__ This paper considers the geometry of the CNN filters. Overparametrized CNN networks are known to exhibit a high level of filter redundancy --- meaning that they are not compact nor information efficient, which results in expensive memory storage requirements. Current approaches to remedy that problem are based on pruning the filters, either in a data dependent manner (seeing which ones are more impactful in training accuracy), or a data-independent way. This paper considers the latter, as this approach is perhaps more likely to give us a better theoretical understanding of the inner workings of CNNs.

To do so, the paper takes its inspiration from archetypal analysis: rather than looking to reduce redundancy by looking at filter centroids, the authors look for ``extreme’’ filters: all filters can be then expressed as a linear combination of these extreme points (or archetypes).  The authors show through a set of experiments that this results in an efficient compression of the neural networks, with no significant loss in representation quality.

__Contributions.__
The paper suggests a new way of assessing filter redundancy ---which suggests a new ranking to prune the network adequately. The paper’s contribution is to frame this problem as archetypal analysis --- the best points are those defining the convex hull of all filters. The algorithm is, however, not novel.



**Questions:**

See weaknesses

**Limitations:**

Given the strengths and weaknesses detailed above.  the paper falls short of its objective. It does not provide enough of a discussion on the benefits of their approach compared to He et al’s (from a computational perspective or theoretical perspective: it could have been interesting to compare for instance, for different metrics, the accuracy loss as a function of pruning percentage). The proposed method is still a heuristic (perhaps better founded than the arbitrary selection of geometric medians, but a heuristic nonetheless). The paper would gain in strength and applicability by leveraging further the robustness results of AA to give more depth to their approach.

**Recommended Decision:**

1: Reject

**Relevance:**

3: Solid fit

**Strengths And Weaknesses:**

__Strengths:__This approach is interesting, as it provides perhaps a little bit more theoretical insights into filter selection than the method by He et al.

__Weaknesses:__

While the approach is quite nice, and draws on Archetypal Analysis (which is really nice and has been showed to be useful in a variety of contexts), this paper falls perhaps a little short of its potential:
+ The authors (contrary to He et al) do not consider the computational complexity of computing the archetypes. While they build on Sharma et al ---whose algorithm has a computational complexity in  O(ndl), with l the number of archetypes----, it would have been nice to see this explicited in the paper.
+ I did not understand the section on filter fabrication: “filters can be fabricated by computing the random convex combinations of these archetypes”. I am not an expert in filter pruning so my background might be lacking, but why do we need to fabricate filters? Can we not simply use the archetypes? If it is indeed shown that convex combinations of filters yield better performance (but why??), what level of redundancy should we aim for? It would be good for the authors to explain a little further this filter fabrication stage, its necessity and relevance.
+ It would also have been interesting to compare a little more theoretically and empirically the pruning achieved by their proposed algorithm and He et al’s. Does it prune the same filters? Can we show that the objectives are the same? Looking at the empirical results, it seems to me that the methods are not statistically significantly different. It would have been great to see a plot or table of the percentage of overlap in the filters chosen for pruning between the two methods.
+ The paper does not try to comment nor to address the potential shortcomings of Archetypal Analysis --- in particular, its sensitivity to outliers. Since the main contribution of this paper is the application of this method to filter pruning and the novelty does not lie in the suggestion of a new method to solve for these, it would have been important to address all the issues that might arise in practice.


**Submission Track:**

Proceedings Paper (9 Page)

---

### Decision · Program_Chairs · 2022-10-21

Accept (Poster)